# Investigation of the lncRNA *THOR* in Mice Highlights the Importance of Noncoding RNAs in Mammalian Male Reproduction

**DOI:** 10.3390/biomedicines9080859

**Published:** 2021-07-22

**Authors:** Lin Zhou, Jianing Li, Jinsong Liu, Anbei Wang, Ying Liu, Hao Yu, Hongsheng Ouyang, Daxin Pang

**Affiliations:** 1Key Lab for Zoonoses Research, Ministry of Education, College of Animal Sciences, Jilin University, Changchun 130062, China; zhoulin19@mails.jlu.edu.cn (L.Z.); lijn17@mails.jlu.edu.cn (J.L.); jsliu17@mails.jlu.edu.cn (J.L.); wangab18@mails.jlu.edu.cn (A.W.); ying19@mails.jlu.edu.cn (Y.L.); yu_hao@jlu.edu.cn (H.Y.); 2Chongqing Research Institute, Jilin University, Chongqing 401123, China

**Keywords:** *THOR*, testis, sperm, MEK-ERK, mice

## Abstract

*THOR* is a highly conserved testis-specific long noncoding RNA (lncRNA). The interaction between *THOR* and the development of the male reproductive system remains unclear. Herein, CRISPR/Cas9 technology was used to establish a stable *THOR*-deficient mouse model, and the relationship between *THOR* and the fertility of adult male mice was investigated. The male mice in which *THOR* was deleted were smaller than the WT male mice. Moreover, their survival rate was reduced by 60%, their fertility was reduced by 50%, their testicular size and sperm motility were reduced by 50%, their testicular cell apoptosis was increased by 7-fold, and their ratio of female-to-male offspring was imbalanced (approximately 1:3). Furthermore, to elucidate the mechanisms of male reproductive system development, the mRNA levels of *THOR* targets were measured by qRT-PCR. Compared with WT mice, the *THOR*-deficient mice exhibited significantly decreased mRNA levels of IGF2BP1, c-MYC, IGF1, and IGF2. MEK-ERK signaling pathway expression was downregulated as determined by Western blot. We found that *THOR* targeted the MER-ERK signaling pathway downstream of IGF2 by binding to IGF2BP1 and affected testicular and sperm development in male mice. These results may also provide perspectives for exploring the roles of lncRNAs in human reproductive development and the pathogenesis and potential therapeutic targets of infertility.

## 1. Introduction

Approximately 75% of the genes in mammalian genomes can be transcribed, but less than 2% have the ability to encode proteins [1,2]. Most transcripts that do not encode proteins are called noncoding RNAs and include short and long noncoding RNAs (lncRNAs) [3,4]. RNAs with more than 200 nt account for most noncoding RNAs and have the most complex functions [5]. LncRNAs were previously described as “junk” genes that do not participate in the mechanisms of individual regulatory networks [6,7]. Recent studies have confirmed that lncRNAs can act as molecular signals, scaffolds, or enhancers/inhibitors to regulate important cell behaviors, including survival, growth, proliferation, and apoptosis resistance [8,9]. The molecular mechanisms of cancer/testis lncRNAs in cancer metastasis are largely unknown and need to be fully elucidated [10].

The lncRNA testis-associated highly conserved oncogenic long non-coding RNA (*THOR*) was the first cancer/testicular lncRNA to be identified and was further shown to have a conserved interaction with insulin-like growth factor 2 mRNA binding protein 1 (IGF2BP1) and to contribute to the mRNA stabilization activity of IGF2BP1. *THOR* deficiency caused fertilization defects in zebrafish and conferred resistance to melanoma onset [11]. The *THOR* represents a novel class of functionally important cancer/testis lncRNAs that have undergone positive evolutionary selection for structure and function [11]. These previous studies have indicated that the *THOR* plays an important role in male reproductive system development.

RNA binding proteins (RBPs) regulate genes via a variety of mechanisms [12]. IGF2BP1 is a member of the conserved single-stranded RBP family, which plays important roles in gene expression regulation [13,14]. The structural characteristics of IGF2BP1 enable it to bind to a variety of target mRNAs, including c-MYC, IGF2, and IGF1, in the context of *THOR* expression regulation [15].

Insulin-like growth factor 2 (*IGF2*) is the first endogenous imprinted gene identified that specifically expresses paternal alleles in most human tissues, while maternal alleles are imprinted [16]. During early embryonic development and sperm development, the *IGF2* expression product IGF is an embryonic growth factor and mitotic peptide that plays an important role in the normal growth and development of embryos [17,18], and IGF2 is an important growth factor during spermatogenesis [19,20].

In the nonclassical regulatory androgen network, the MEK and ERK kinases participate in the MAP kinase phosphorylation cascade [21,22]. MEK can activate ERK kinase and affect the existence and development of normal sperm in the testis [23], and the MEK-ERK signaling pathway is downstream of IGF2 [24]. Numerous studies have shown that *THOR* stabilizes the expression of its downstream mRNA targets by binding IGF2BP1 [25,26], and *THOR* expression was positively correlated with the mRNA expression of IGF2BP1 [27]. However, the relationship of *THOR* with IGF2BP1 during testis and sperm development is unclear.

We successfully constructed *THOR*-deficient mice using CRISPR/Cas9 technology and found that the expression levels of *THOR* and IGF2BP1 were decreased in testicular tissues, that testicular development was defective, and that sperm maturation was decreased significantly. In addition, *THOR* regulated the downstream IGF2BP1-IGF2-MEK-ERK signaling pathway together with IGF2BP1, which affected testicular and sperm development. Our findings further elucidate the relationship between lncRNAs and reproductive system development and provide ideas for the treatment of male infertility.

## 2. Materials and Methods

### 2.1. Animals and Ethical Statement 

All mouse experiments in this study were approved by the animal protection and research ethics committee of Jilin University, and all procedures were strictly in accordance with the guidelines for the care and use of laboratory animals. Specific pathogen-free C57/BL6 mice were purchased from Liaoning Changsheng Biotechnology Co., Ltd. (Benxi, China) and raised at the experimental animal center of Jilin University. All animal operations were performed under anesthesia, and every effort was made to minimize pain.

### 2.2. CRISPR/Cas9 sgRNA Preparation, Embryo Microinjection, and Embryo Transfer 

The CRISPR/Cas9 sgRNA was designed (http://crispr.mit.edu/ accessed on 1 July 2021) and assembled as previously described [28]. The annealed sgRNA oligos were cloned into the BbsI sites of the pUC57-T7-sgRNA cloning vector (Addgene ID 51306) as previously described [29]. The pUC57-T7-sgRNA vector was PCR amplified using T7 primers (T7-F: 5′-GAAATTAATACGACTCACTATA-3′ and T7-R: 5′-AAAAAAAGCACCGACTCGGTGCCAC-3′), and the PCR products were transcribed in vitro with the MAXIscript T7 Kit (Ambion, Austin, TX, USA) and purified with the miRNeasy Mini Kit (Qiagen, Dusseldorf, Germany) in accordance with the manufacturers’ instructions. The 3xFLAG-NLS-SpCas9-NLS vector (Addgene ID 48137) was linearized with NotI (NEB, Beijing, China) and transcribed in vitro using the mMessage mMachine SP6 Kit (Ambion, Austin, TX, USA) and the RNeasy Mini Kit (Qiagen, Dusseldorf, Germany) in accordance with the manufacturers’ instructions. The microinjection and embryo transfer procedures were performed essentially as we previously described [30].

### 2.3. Detection of THOR Deletion in Mouse Pups by PCR and Sequencing 

Genomic DNA was extracted from small pieces of tail tissues from *THOR*-deficient and WT mice using the TIANamp Genomic DNA Kit (Tiangen, Beijing, China) in accordance with the manufacturer’s instructions. The sgRNA target sites were amplified by PCR using primers shown in Appendix A. The PCR product was gel purified and cloned into the pGM-T vector (Tiangen, Beijing, China), and DNAman was then used for sequencing and analysis.

### 2.4. T7E1 Cleavage Assay 

The T7E1 cleavage assay was performed as described previously [31]. Briefly, the PCR products were purified, denatured, and then reannealed in NEBuffer 2 (NEB, Beijing, China) using a thermocycler. Hybridized PCR products were digested with T7 endonuclease 1 (NEB, Beijing, China, M0302L) for 30 min at 37 °C and subjected to 2% agarose gel electrophoresis.

### 2.5. Off-Target Analysis 

The top off-target sites were predicted using the online CRISPR Design tool developed by the Zhang group at the Massachusetts Institute of Technology (http://crispr.mit.edu/ accessed on 1 July 2021). The PCR products of these potential off-target sites obtained using the primers listed in Appendix A were subjected to the T7E1 assay and Sanger sequencing.

### 2.6. RNA Isolation and qRT-PCR 

Total RNA was isolated from the testes of WT and *THOR*-deficient mice using TRIzol-A^+^ reagent (Tiangen, Beijing, China) and treated with DNase I (Fermentas, Burlington, Ontario, Canada, EN0525). First-strand complementary DNA (cDNA) was synthesized using a cDNA first-strand synthesis kit (Tiangen, Beijing, China) and then subjected to qRT-PCR to examine the expression of *THOR*. The primers used for qRT-PCR are shown in Appendix A. qRT-PCR was performed using the BioEasy SYBR Green I Real Time PCR Kit (Bioer Technology, Hangzhou, China), and relative gene expression levels were determined by the 2-ΔΔCT method formula. *GAPDH* was used as the reference gene. All experiments were repeated three times, and the data were expressed as the mean ± SD.

### 2.7. Body Weight, Food Intake, Survival, and Statistical Analyses 

The body weights and food intake of age- and sex-matched WT and *THOR*-deficient mice were measured weekly. All data were expressed as the mean ± SD., and a minimum of three individual animals of each genotype was used in all experiments. The data were analyzed by Student’s *t*-test using GraphPad Prism 8.0 software. A probability of *p* < 0.05 was considered statistically significant.

### 2.8. Hormone Level Analysis 

In total, 200 µL of blood was harvested from the tail tip of 8-week-old *THOR*-deficient and WT mice and centrifuged at 1000 rpm/min at 4 °C for 10 min to collect the serum (at least 3 mice per group). The T and LH hormone levels in the serum samples were determined using ELISA kits (NJJCBIO, Nanjing, China). A probability of *p* < 0.05 was considered statistically significant.

### 2.9. Histological Analysis 

Various tissues, including testicular, epididymal, vas deferens, seminal vesicle, and bladder tissues, were collected from *THOR*-deficient and WT mice (euthanized at 8 weeks of age). The tissues were fixed in 4% paraformaldehyde at 4 °C, dehydrated in increasing concentrations of ethanol (70% for 6 h, 80% for 1 h, 96% for 1 h, and 100% for 3 h), cleared in xylene, and embedded in paraffin for histological examination. The 5-μm-thick sections were cut for H&E and TUNEL staining as previously described. The stained sections were imaged with a Nikon TS100 microscope.

### 2.10. Morphological Analysis of Testes 

H&E-stained cross-sections of testes from 8-week-old *THOR*-deficient and WT mice were analyzed for seminiferous tube size. A minimum of three different regions were analyzed per section. The seminiferous tube size and cross-sectional area were calculated using ImagePro Plus 6.0 software (Media Cybernetics, Silver Spring, MD, USA). Apoptosis analysis was performed on TUNEL-stained testicular cross-sections from 8-week-old *THOR*-deficient and WT mice. A minimum of three different regions were analyzed per section. The number of apoptotic seminiferous cells was calculated using ImagePro Plus 6.0 software (Media Cybernetics, Silver Spring, MD, USA).

### 2.11. Detection and Analysis of Offspring Fertility 

Pairs of *THOR*-deficient and WT mice (*THOR*-deficient and *THOR*-deficient; *THOR*-deficient and WT; 1:1 female-to-male ratio; three pairs of repeats in each group) were housed together in cages for approximately 6 months. The fertility and female-to-male ratios of the mice were compared. After the *THOR*-deficient and WT mice were divided into cages, the embryos of female mice with vaginal plugs were collected, and the fetuses were cultured in vitro to develop blastocysts. At least three pairs were analyzed in each group.

### 2.12. Embryonic Development and Statistical Analyses 

WT fertilized eggs were cytoplasmically injected to delete *THOR*, and embryonic development in the *THOR*-deficient mice was examined. The experiments were repeated three times, and each group had approximately 100 fertilized eggs. The statistical results were analyzed with GraphPad Prism 8.0. A probability of *p* < 0.05 was considered statistically significant.

### 2.13. Sperm Motility and Morphological Analysis 

Sperm were taken from the epididymal tails of 8-week-old male *THOR*-deficient and WT mice and suspended in HTF medium (Sigma, St. Louis, MO, USA). The sperm were counted on a blood cell counting plate under a Nikon TS100 microscope. The sperm suspension was used to make a sperm smear, and eosin-nigrosin staining was performed to observe deformity under a microscope.

### 2.14. In Vitro Fertilization (IVF) 

Superovulation was induced in C57BL/6 female mice (4 weeks old) via intraperitoneal injections of pregnant mare serum gonadotrophin (PMSG) on day 1 and human chorionic gonadotropin (hCG) on day 3. On day 4, male mice were sacrificed, and the cauda epididymis was quickly removed, allowing the sperm capacitates to be incubated for 1 h prior to insemination. During this period, the superovulated female mouse was sacrificed, and the ampullae were quickly removed and placed in a fertilization dish (FERTIUP PM 0.5 ML-CARD MEDIUM set, KYD-005-EX) to collect the cumulus-oocyte complexes (COCs). The sperm were added to the fertilization dish including the COCs or cumulus-free oocytes using a pipette, and dish was incubated for approximately 3 h at 37 °C. The embryos were then removed from the fertilization dish and washed 3 times with HTF. The KSOM drops covered with paraffin oil were incubated overnight at 37 °C, and the number of embryos in the two-cell stage was determined on day 5.

### 2.15. Protein Isolation and Western Blotting 

Total protein was extracted from tissues using radioimmunoprecipitation assay (RIPA) lysis buffer supplemented with phenylmethanesulfonyl fluoride (PMSF, Roche Applied Science, Basel, Switzerland) and phosphatase inhibitor (PI, Thermo Scientific, Waltham, MA, USA). The protein concentration was measured using an enhanced BCA protein assay kit (Beyotime, P0010, Shanghai, China). Total protein extracts were separated on 10% gels via SDS-PAGE and then transferred to 0.22 nm polyvinylidene fluoride membranes (Millipore, Burlington, MA, USA). The proteins were probed with specific antibodies after the blot was blocked with 5% non-fat milk (Boster, AR0104, Wuhan, China). The antibodies used in this study are listed in Appendix A.

### 2.16. Statistical Analysis 

Significant differences were determined via a two-tailed unpaired Student’s *t*-test or analysis of variance using GraphPad Prism 8.0. The data were expressed as the mean ± SD. * *p* < 0.05, ** *p* < 0.01, *** *p* < 0.001 and **** *p* < 0.0001 denoted the significance thresholds; ns denoted not significant.

## 3. Results

### 3.1. Evolutionary Relationship and Tissue Conservation of THOR in Vertebrates

We aligned the human *THOR* sequence with 11 representative orthologous sequences using BLAST and revealed high conversation in vertebrates. The neighbor-joining tree further showed that *THOR* first originated in fish and that mouse *THOR* was most closely related to primate species (humans, macaques, baboons) (Figure 1A). Next, we predicted regions that are homologous to the human *THOR* in the genomes of 4 commonly used experimental animals (mice, pigs, rabbits, and zebrafish) and identified 2 exon homologues. We thus showed that *THOR* is highly conserved in various species (Appendix A). Quantitative real-time PCR (qRT-PCR) was used to detect *THOR* expression in 24 tissues and organs of adult mice, further revealing that *THOR* has high testicular specificity. The expression trends were consistent with those reported in a previous study (Figure 1B). In vertebrates, the testicular expression of *THOR* was highly conserved and had a substantial impact on male reproduction.

### 3.2. Generation of THOR-Deficient Mice 

The CRISPR/Cas9 platform was used to edit the *THOR* mouse gene (Figure 2C) Because *THOR* has two exons, cannot be translated into a protein, and has non-defined introns [11], to assess the involvement of the gene regulatory network, sgRNAs were designed to delete large fragments of *THOR* that included both exonic and intronic regions (Figure 2A,B). This method was equivalent to the complete destruction of the gene *THOR* structure. We injected the in vitro-transcribed Cas9 mRNA and sgRNA mixture into the fertilized egg by cytoplasmic injection and then transplanted the embryo into a pregnant mouse to obtain an F0 generation mouse, and the identities of positive mouse genomes were verified (Figure 2D,E). Breeding of the offspring obtained *THOR*-deficient mice (Figure 2C), which was further verified (Figure 2F). We also used qRT-PCR primers located outside the gene deletion region to detect the decreased expression of *THOR* (Appendix A). It shows that after the large fragment of *THOR* deficient, the gene recombines to form a new truncated *THOR*. Off-target effects are a major problem limiting the practical applications of CRISPR/Cas9 technology. To assess potential off-target effects in *THOR*-deficient mice, 10 pairs of PCR primers were designed based on potential off-target sites (Appendix A). PCR sequencing and the T7E1 cleavage assay for the detection of mutations revealed no off-target effects in the samples (Figure 2G). We further analyzed and compared the secondary structures of the mouse *THOR* gene and found that the secondary structure of *THOR* was completely changed after *THOR* gene deletion (Figure 2H, http://rna.tbi.univie.ac.at/cgi-bin/RNAWebSuite/SCA.cgi accessed on 1 July 2021) and that IGF2BP1 and downstream signaling regulatory pathways together had a substantial impact.

### 3.3. THOR-Deficient Mice Exhibit Decreased Fertility and a Female-to-Male Imbalance

*THOR*-deficient mice were housed in the same cages as WT mice during the breeding process, and the *THOR*^+/−^ and *THOR*^−/−^ and WT mice were weighed regularly. *THOR* was shown to regulate the body weights of both groups of mice, and *THOR*^−/−^ mice weighed less than *THOR*^+/−^ mice (Figure 3A). To determine whether this differential result was due to the influence of *THOR* on mouse food intake, we statistically analyzed the food intake of *THOR* on WT and *THOR*-deficient mice. No significant difference in food intake was observed between WT and *THOR*-deficient mice (Figure 3B); thus, the effect of *THOR* on mouse body weight was presumably caused by genetic factors. To determine whether *THOR* affects mouse survival, we assessed the survival of WT, *THOR*^+/−^, and *THOR*^−/−^ mice for 126 days (Figure 3C) and found that the survival rate of *THOR*-deficient mice was lower than that of WT mice and that the *THOR*-deficient mice were more likely to die.

*THOR* is specifically and highly expressed in the testes. To explore the effects of *THOR* on the mouse reproductive system, we compared the testicular weights and testicular mass indices of *THOR*-deficient and WT mice (Figure 3D). The testes of *THOR*-deficient mice were smaller and lighter than those of WT mice, the testes of *THOR*^−/−^ mice were smaller than those of *THOR*^+/−^ mice, and the testes mass indices of *THOR*^−/−^ mice were lower than those of WT mice. Therefore, we proved that *THOR* affects testicular development in mice.

To determine which aspects of *THOR* affect testicular development in male mice, we assessed the fertility of the mice. In vitro fertilization (IVF) experiments were performed using WT and *THOR*^+/−^ and *THOR*^−/−^ mouse sperm and WT oospheres, with an average of 100 oospheres in each group. The experiments were repeated 3 times at 37 °C in a 5% CO_2_ incubator for 12 h, and embryonic development was observed. The 2-cell stage served as the marker for oosphere detection (Figure 3E). We found that the fertilization rate of WT mouse sperm was above 90%, while that of *THOR*^+/−^ mouse sperm was approximately 72%, and that of *THOR*^−/−^ mouse sperm was approximately 50%. Therefore, we hypothesize that *THOR* is involved in sperm development and maturation.

We microinjected 100 oospheres and deleted the mouse *THOR* gene using CRISPR/Cas9 technology. After microinjection, the oospheres were developed to the blastocyst stage in vitro to observe embryonic development (Figure 3F). Deletion of *THOR* by microinjection lowered the embryo development rate in vitro compared to that in the control group, and the blastocyst development rate decreased from an average of 36% to approximately 25%. Thus, *THOR* affects the growth and development of embryos.

We next assessed the effect of *THOR* on mouse fertility. We mated female and male 8-week-old mice at a 1:1 ratio during the estrus period, with 6 pairs of mice in each group. The vaginas of the female mice were checked at 8 a.m. the next morning. One hundred percent of the female and male WT mice successfully mated, while *THOR* had a slight impact on the fertility of the female mice, and the genotypes of the *THOR*^+/−^ and *THOR*^−/−^ female mice were not obviously different. The fertility of *THOR*-deficient male mice was reduced by 50% (Figure 3G). Therefore, we believe that *THOR*, as a highly conserved testis-specific lncRNA, plays an important role in the regulatory network of the reproductive system in male mice. After evaluating the vaginal suppositories, we observed female mice implanted with embryos, and the 2-cell embryo stage on the second day indicated fertilization. We found that *THOR* affected not only the mating conditions of the male mice, but also the combination of sperm and egg cells (Figure 3H). The embryo fertilization rate was relatively low, and the *THOR*^−/−^ males were more seriously impacted than the *THOR*^+/−^ males. Therefore, we hypothesize that *THOR* affects testicular and sperm development and that decreased male fertility is due to the absence of *THOR*.

We next assessed whether *THOR* affects mouse fertility from the perspective of individual mouse reproduction. We mated 6-week-old female and male mice of different genotypes at a 1:1 ratio to assess the reproduction of the offspring within 6 months (Figure 3I, Appendix A). The deletion of *THOR* in male and female mice decreased the number of litters produced, which affected the offspring ratio of female- to-male (Figure 3J). Under natural circumstances, the female-to-male ratio of WT mouse offspring is approximately 1:1. Deletion of the *THOR* gene in female mice had little effect on the number of female offspring. However, deletion of *THOR* in male mice substantially increased the number of male offspring, and the female-to-male ratio was approximately 1:3. Therefore, we hypothesized that *THOR* deficiency impacted the fertilization ability of X and Y sperms, thereby resulting in more male offspring.

### 3.4. THOR Deficiency Affects Testicular Development in Male Mice

To determine the effect of *THOR* on testicular development in male mice, we compared photos of adult male mice at 8 weeks of age (Figure 4A). The reproductive systems of WT mice developed normally, while the *THOR*^+/−^ and *THOR*^−/−^ mice exhibited substantially fewer sperm in the seminal vesicles than the WT mice and low numbers of mature sperm (Figure 4B). In addition, the testes were not fully developed and differed in size. The testes of *THOR*^+/−^ and *THOR*^−/−^ mice were smaller than those of WT mice, and the testes of *THOR*^−/−^ mice were smaller than those of *THOR*^+/−^ mice (Figure 4C).

We paraffin-embedded mouse testes sections and performed hematoxylin and eosin (H&E) (Figure 4D) and TUNEL (Figure 4E) staining. In the testes of *THOR*-deficient mice, the diameters of seminiferous tubules were decreased and sparsely arranged and Sertoli and germ cells were arranged abnormally (Figure 4F). TUNEL staining revealed significantly increased apoptosis levels in *THOR*^+/−^ and *THOR*^−/−^ mice (Figure 4G). Statistical analysis revealed that the apoptosis index was increased in the absence of *THOR*.

Therefore, we conclude that the highly conserved *THOR* gene, which is specifically expressed in the testis, not only participates in the growth and proliferation of numerous tumor and cancer cells but also plays an important role in the regulation of the male reproductive system; however, the specific mechanism is currently unknown.

### 3.5. THOR Deficiency Affects Sperm Development

To determine whether *THOR* affects sperm development in male mice, we counted the sperm of male WT, *THOR*^+/−^, and *THOR*^−/−^ mice (3 male mice of each genotype were analyzed). The epididymal tails on both sides of each sperm were digested in 1 mL of HTF culture medium. After the sperm swam out, the sperm density, motility, and deformity rate were determined. Deletion of *THOR* reduced the sperm density and motility in male mice; the sperm density and motility of *THOR*^−/−^ mice were lower than those of *THOR*^+/−^ mice, while the deformity rate was higher for *THOR*^−/−^ mice (Figure 5A). These data suggest that *THOR* affects the development and maturation of sperm in male mice.

Previous studies have shown that hormones are regulated within the developmental receptors of male sperm. While *THOR* clearly affected testicular and sperm development, its effects on bodily hormone levels were unknown. Therefore, we assessed the serum testosterone (T) and luteinizing hormone (LH) levels in 8-week-old male WT, *THOR*^+/−^, and *THOR*^−/−^ mice (6 mice in each group). The serum T and LH levels of *THOR*-deficient mice were 50% lower than those of WT mice. The T and LH levels in *THOR*^−/−^ mice were lower than those in *THOR*^+/−^ mice (Figure 5B). Therefore, we believe that *THOR* effects the expression of sex hormones in male mice and helps to regulate testicular and sperm development.

Sperm were morphologically assessed by eosin-nigrosin staining (Figure 5C). *THOR*-deficient mice exhibited sperm with ten different features: large head, small head, no head, irregular head, no hook, neck curvature, double body, body curl, curly tail, and short tail (Figure 5D). The deformity rate of *THOR*-deficient sperm was higher than that of WT sperm. Therefore, we conclude that *THOR* participates in a certain regulatory mechanism that affects the development and maturation of sperm.

### 3.6. THOR Affects Sperm Development through the MEK Pathway

IGF2BP1 is an RBP that mediates RNA stability and promotes the translation of numerous well-defined mRNA targets by binding to RNA and other RBPs [32,33]. Many previous studies have shown that *THOR* specifically binds to IGF2BP1 in the nucleus and regulates the stability of mRNAs, such as IGF1, IGF2, and c-MYC [34]. Moreover, IGF2 has been shown to effect testicular development and spermatogenesis [35]. Therefore, we speculate that *THOR* and IGF2BP1 together effect testicular and sperm development in males by regulating the downstream IGF2 signaling pathway. To further confirm the above hypothesis, we assessed the IGF2BP1-IGF2-MEK-ERK signaling pathway in the testicular tissues of WT and *THOR*-deficient mice by qRT-PCR (Figure 6A) and verified the results by Western blot (Figure 6B). The Western blot protein bands were semiquantitatively analyzed by ImageJ, revealing that the deletion of the *THOR* affected the regulation of the MEK-ERK pathway (Figure 6C). Overall, these data indicate that *THOR* mediates the stabilization of IGF2BP1 and participates in the IGF2BP1 and IGF2 coregulation of the MEK signaling pathway and in testicular and sperm development (Figure 6D).

## 4. Discussion

RNAs with a fragment length of more than 200 nt and noncoding ability are called lncRNAs, which have complex physiological functions [36,37]. The lncRNA *THOR* was the first cancer/testicular lncRNA to be discovered and was further shown to conservatively interact with IGF2BP1 and to contribute to its mRNA stabilization activity [11]. *THOR* regulates the proliferation and development of cancer cells by affecting the expression of the IGF2BP1 downstream signaling pathway and has been shown to play a carcinogenic role in liver cancer, gastric cancer, renal cell carcinoma, osteosarcoma, nasopharyngeal carcinoma, tongue squamous cell carcinoma, colorectal cancer, melanoma, triple negative breast cancer, and non-small cell lung adenocarcinoma [38,39]. Notably, transgenic *THOR* knockout produced fertilization defects in zebrafish and conferred resistance to melanoma onset [11]. While these results indicate that *THOR* plays an important role in male reproductive system development, little is known about its role in mammalian testis and sperm development.

Xue and Zhong et al. confirmed that the interaction between *THOR* and IGF2BP1 is conserved among many vertebrate species [38]. We explored whether similar effects occur in mammals and used CRISPR/Cas9 technology to construct a mouse model of *THOR* gene deletion (2613 bp). In order to detect the expression of *THOR*, we used qRT-PCR primers located in the gene deletion region, the expression of *THOR* is lost. We also used qRT-PCR primers located outside the gene deletion region to detect the decreased expression of *THOR*, but not the original *THOR*. It was proved that after the large fragment of *THOR* was deleted, the gene is recombined to form a new truncated *THOR*. Therefore, when we are studying the function of lncRNAs, we can delete a large fragment of lncRNAs to cause recombination of the remaining fragments, structural changes, and produce a phenotype similar to the complete deletion of the entire lncRNAs. The *THOR*-deficient male mice were smaller than WT mice; moreover, their survival rate was reduced by 60%, their fertility was reduced by 50%, and their female-to-male offspring ratio was approximately 1:3. Further analysis of the effect of *THOR* on male reproductive system development showed that the testicular size and sperm motility of *THOR*-deficient male mice were reduced by 50%, that the diameter of seminiferous tubules was decreased, and that the apoptosis of testicular cells was increased by 7-fold, which confirmed that *THOR* is involved male testicular development. We next investigated whether *THOR* plays an important role in sperm development and found that the sperm density in the epididymal tissues of *THOR*-deficient male mouse was decreased by 50%, that the sperm motility was decreased, that the sperm deformity rate was increased, and that the serum T and LH levels were decreased by 50%, thereby indicating that deletion of the *THOR* gene affected male sperm development. In conclusion, the lncRNA *THOR* plays an important role in male reproductive system development.

As a signal transduction protein, ERK can transmit mitogen signals. Under normal conditions, ERK is localized in the cytoplasm and can be activated by various growth factors, peroxides, and free radicals [40]. MEK is upstream of ERK and participates in the process by which ERK enters the nucleus [41]. MEK-ERK is the downstream signaling pathway of IGF2 [42]. To prove that the MEK-ERK pathway is involved in testicular and sperm development, we analyzed the mRNA levels of the downstream target genes of *THOR* by qRT-PCR and revealed that the levels of IGF2BP1, c-MYC, IGF1, and IGF2 were significantly decreased in *THOR*-deficient mice. Further verification by WB showed reduced expression of the MEK-ERK pathway was. Our results indicate that *THOR* regulates the MEK-ERK signaling pathway downstream of IGF2 by binding to IGF2BP1, regulates testicular and sperm development, and affects the viability, fertility, and sex ratio of offspring in male mice.

The lncRNA *THOR* is known to be expressed in a variety of species and human cancers, which furthers our understanding of the correlation between normal and disease processes [11]. Deletion of *THOR* in zebrafish can lead to abnormal fertilization, and we found that the deletion of *THOR* in male mice affected their survival and fertility as well as the sex ratio of their offspring. In addition, the regulatory effects of *THOR* on male reproductive system have not been fully elucidated. We further explored the influence of *THOR* on the female-to-male offspring ratio, but whether *THOR* specifically regulates the reduced vitality and energy of X sperm remains unclear.

We herein used CRISPR/Cas9 technology to construct a *THOR*-deficient mouse model to assess the effects of *THOR* on testicular and sperm development in male mice. The results of this study provide new perspectives for exploring reproduction-specific lncRNAs, furthers our understanding of the roles of lncRNAs in human reproduction and development, and provides more information on the pathogenesis of infertility and potential therapeutic targets.

## Figures and Tables

**Figure 1 biomedicines-09-00859-f001:**
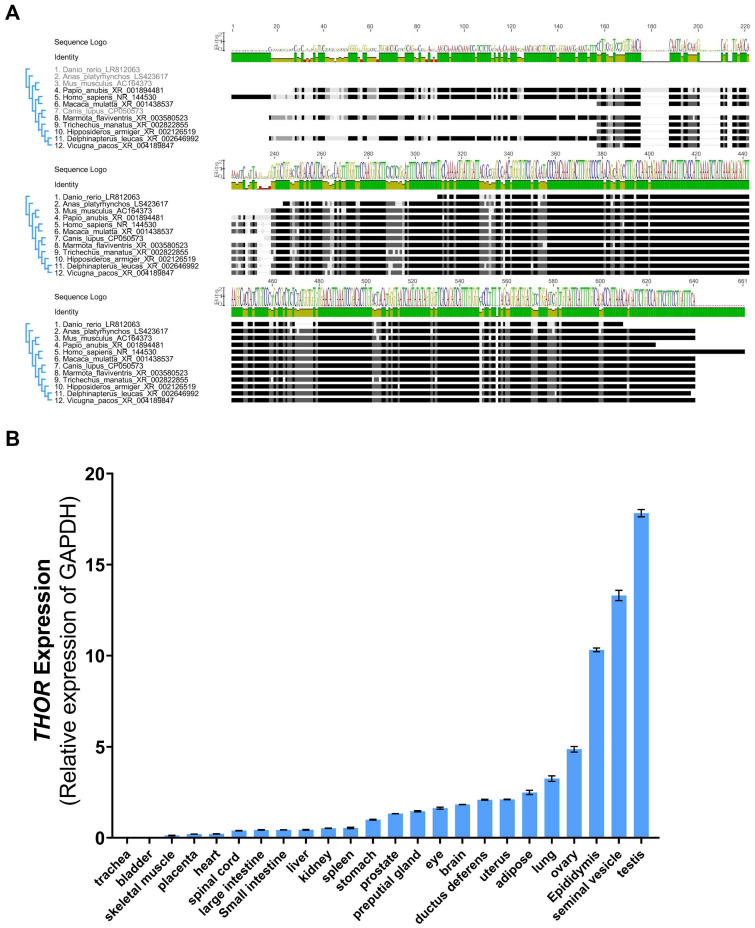
The evolutionary relationship and tissue conservation of *THOR* in vertebrates. (**A**) Sequence alignment of *THOR* in 12 representative vertebrates. (**B**) Measurement of mouse *THOR* expression by qRT-PCR in an adult murine tissue panel (data are shown as the mean ± SD from two independent experiments).

**Figure 2 biomedicines-09-00859-f002:**
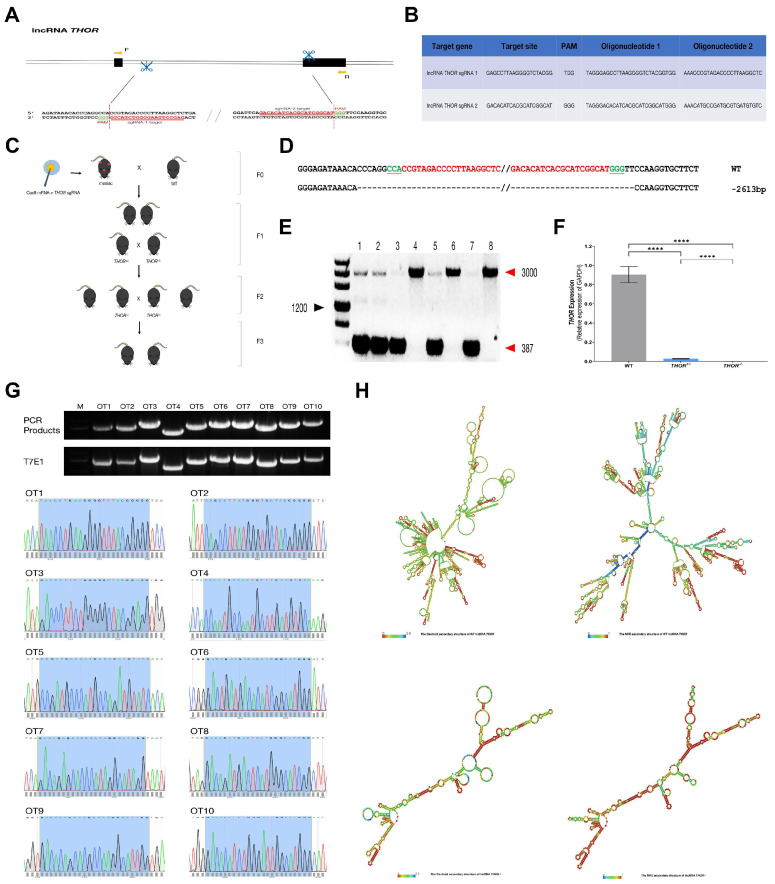
Generation of lncRNA *THOR*-deficient mice. (**A**) Schematic diagram of the sgRNA targeting of the mouse *THOR* gene locus. *THOR* exons are indicated by rectangles, the target sites of the 2 sgRNA sequences (sgRNA 1 and sgRNA 2) are highlighted in red, and the protospacer-adjacent motif (PAM) sequence is highlighted in green. Primers F and R were used to detect mutations in mice. (**B**) Target sequences of the two sgRNAs and complementary oligo sgRNAs. (**C**) Schematic depicting the creation of the *THOR*-deficient mouse model. (**D**) *THOR* mutation detection in mice by T-cloning and Sanger sequencing. (**E**) PCR analysis of *THOR* mutations in pups (M) D2000; 1–8 represent the newborn mice used in this study. (**F**) Mutations in 8-week-old *THOR*^+/−^ and *THOR*^−/−^ were detected by qRT-PCR. Compared with that in the WT control group, the expression of *THOR* in *THOR*-deficient mice was significantly decreased. **** indicates *p* ≤ 0.0001 as determined by a *t*-test. The data are shown as the mean ± SD from two independent experiments. (**G**) Detection of off-target effects in *THOR*-deficient mice. T7E1 cleavage analysis of three potential off-target sites (POTS) for sgRNA 1, M, DL2000; 1–3 represent the three POTS. T7E1 cleavage analysis of seven POTS for sgRNA 2, M, DL2000; 4–10 represent the seven POTS. T-cloning and Sanger sequencing of the three POTS for sgRNA1. The region including the 20 bp of the POTS and the PAM is shown in the shadow. T-cloning and Sanger sequencing of the seven POTS for sgRNA 2. The region including the 20 bp of the POTS and the PAM is shown in blue. (**H**) The centroid and MFE secondary structures of *THOR* and *THOR*^−/−^ mice were predicted (http://rna.tbi.univie.ac.at/ accessed on 1 July 2021). The red color indicates strong confidence for the prediction of each base.

**Figure 3 biomedicines-09-00859-f003:**
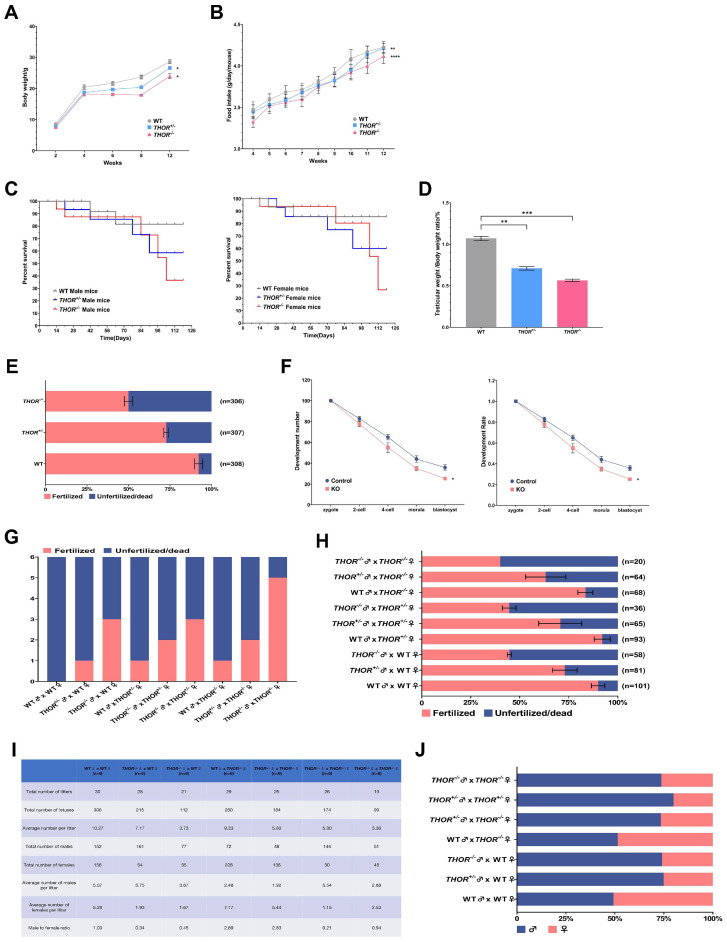
*THOR*-deficient mice exhibit decreased fertility and an imbalance between females and males. (**A**) Comparison of the body weights of WT and *THOR*-deficient male mice. * indicates *p* ≤ 0.05 as determined by a *t*-test. The data are shown as the mean ± SD from two independent experiments. (**B**) Food intake changes in the mice. ** indicates *p* ≤ 0.01 as determined by a *t*-test, and **** indicates *p* ≤ 0.0001 as determined by a *t*-test. The data are shown as the mean ± SD from two independent experiments. (**C**) Survival curve statistics of WT and *THOR*-deficient mice. Ten mice in each group. (**D**) The testes/body weights of WT and *THOR*-deficient mice. ** indicates *p* ≤ 0.01 as determined by a *t*-test, and *** indicates *p* < 0.001 as determined by a *t*-test. The data are shown as the mean ± SD from two independent experiments. (**E**) The fertilization rates of WT female mouse oocytes after in vitro fertilization with sperm from WT and *THOR*-deficient male mice. (**F**) Embryonic development after *THOR* deletion by microinjection of the sgRNA and Cas9 mRNA mixture. Each group of 100 oospheres, repeat 3 times. * indicates *p* ≤ 0.05 as determined by a *t*-test. The data are shown as the mean ± SD from two independent experiments. (**G**) Female mice in estrus were mated with male mice of different genotypes, and the vaginal suppositories were counted the next morning. Six pairs of mice in each group. (**H**) Female mice in estrus were mated with male mice of different genotypes, and the fertilization of female mice with vaginal suppositories was assessed the next morning. n is the total number of embryos in female mice whose vaginal plugs were detected after mating in Figure 3G. Cultured in vitro for 24 h developed into 2-cell as a fertilization mark. (**I**) The reproductive statuses of the offspring of mice with different genotypes after 6 months of mating at a 1:1 ratio. (**J**) The female-to-male ratio of offspring from mice of different genotypes after 6 months of mating at a 1:1 ratio.

**Figure 4 biomedicines-09-00859-f004:**
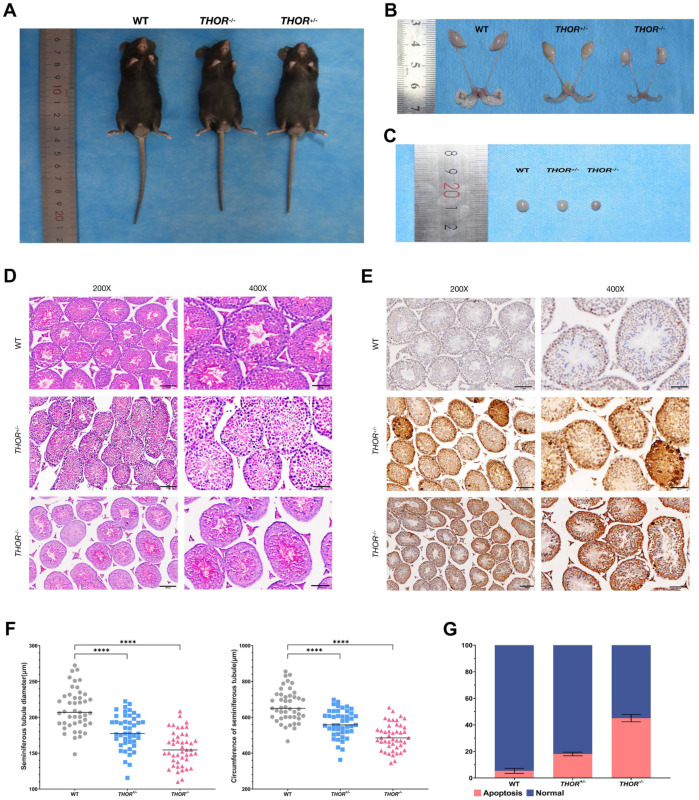
*THOR* deficiency affects testicular development in male mice. (**A**) Photos of 8-week-old male WT and *THOR*-deficient mice. From left to right are WT, *THOR*^−*/*^^−^, *THOR*^+*/*^^−^ mice. (**B**) Comparison of the reproductive systems of 8-week-old male WT and *THOR*-deficient mice. From left to right are WT, *THOR*^+*/*^^−^, *THOR*^−*/*^^−^ mice. (**C**) Comparison of the testicular morphologies of 8-week-old WT and *THOR*-deficient mice. From left to right are WT, *THOR*^+*/*^^−^, *THOR*^−*/*^^−^ mice. (**D**) H&E-stained tracheal sections of the testes of WT mice and *THOR*-deficient mice. In the testes of *THOR*-deficient mice, the seminiferous tubules had decreased diameters and were sparsely arranged and Sertoli and germ cells were arranged abnormally. (**E**) Immunohistochemical staining of the testes of WT and *THOR*-deficient mice. (**F**) Comparison of the diameters and circumferences of WT and *THOR*-deficient mouse seminiferous tubules. **** indicates *p* ≤ 0.0001 as determined by a *t-*test. The data are shown as the mean ± SD from two independent experiments. (**G**) The apoptotic rates of Sertoli cells in the seminiferous tubules of WT and *THOR*-deficient mice were determined by immunohistochemistry.

**Figure 5 biomedicines-09-00859-f005:**
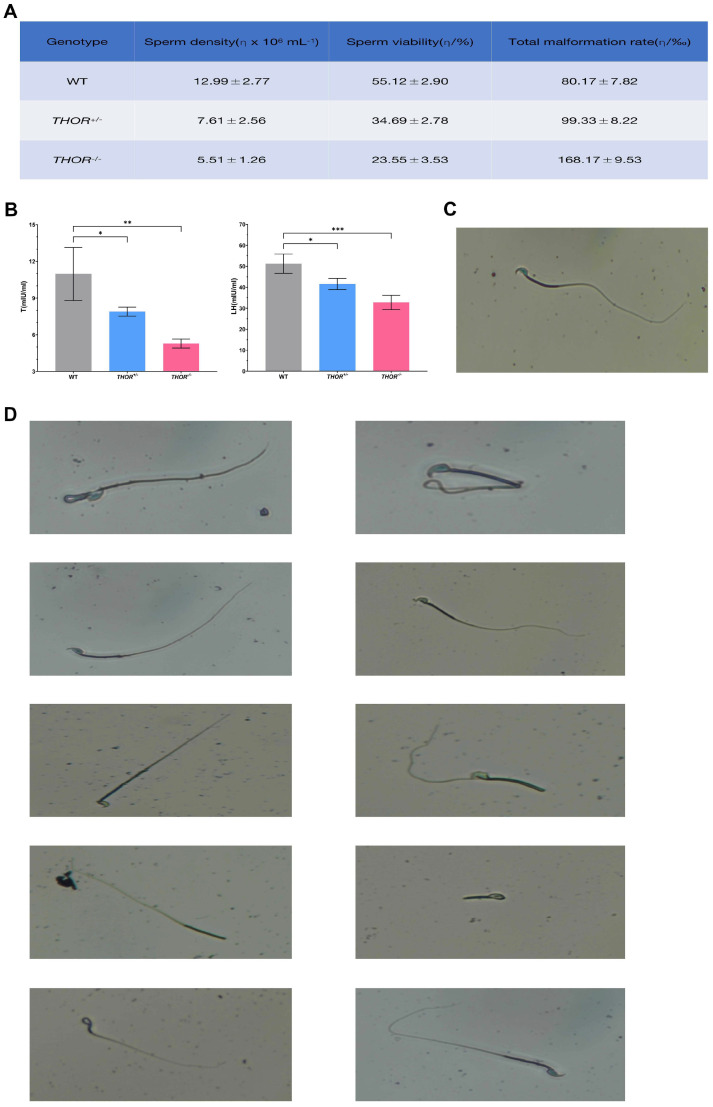
*THOR* deficiency affects sperm development. (**A**) Comparison of the density, motility, and deformity rates of epididymal sperm in WT and *THOR*-deficient mice. (**B**) Comparison of the levels of T and LH in the blood of WT and *THOR*-deficient mice. * indicates *p* ≤ 0.05 as determined by a *t*-test, ** indicates *p* ≤ 0.01 as determined by a *t*-test, and *** indicates *p* ≤ 0.001 as determined by a *t*-test. The data are shown as the mean ± SD from two independent experiments. (**C**) Normal sperm stained with eosin-nigrosin. (**D**) Abnormal sperm from *THOR*-deficient mice stained with eosin-nigrosin.

**Figure 6 biomedicines-09-00859-f006:**
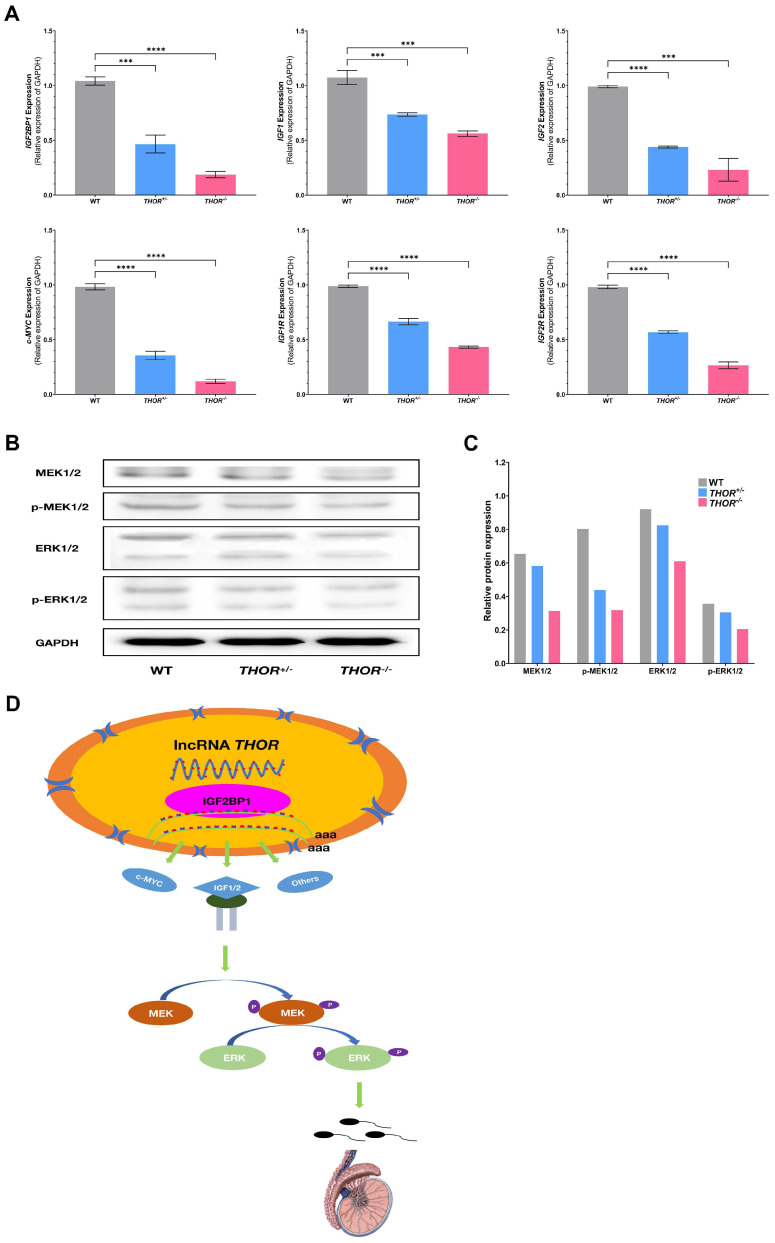
*THOR* deficiency affects sperm development. (**A**) Effect of *THOR* deficiency on the relative mRNA expression levels of target genes as determined by qRT-PCR. *** indicates *p* ≤ 0.001 as determined by a *t*-test, and **** indicates *p* < 0.0001 as determined by a *t-*test. The data are shown as the mean ± SD from three independent experiments. (**B**) Western blot analysis of testicular tissues from *THOR*-deficient mice. (**C**) Semiquantitative analysis of protein expression. (**D**) The mechanism of the effect of *THOR* on testicular and sperm development.

## Data Availability

All data generated or analyzed during this study are included in this published article.

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
