# Peer review of "Investigation of the lncRNA THOR in Mice Highlights the Importance of Noncoding RNAs in Mammalian Male Reproduction"

_biomedicines, 2021, doi:10.3390/biomedicines9080859_

Round 1

Reviewer 1 Report

The manuscript "Investigation of the lncRNA THOR in mice highlights the im-
portance of noncoding RNAs in mammalian male reproduction" by Zhou et al. presents the phenotypic description of a genetically modified mouse line that was generated using CRISR/CAS9 technology. The target is a two exon long non-coding RNA named THOR. The naming THOR stems from its know features of being testes associated.

I think the manuscript describes very strong phenotypes. As I am not a molecular biology expert, these in principle seems appropriate. My main questions would derive from the mthod itself. Here CRISPR/CAS9 was applied to a two exon non-coding RNA. IN Figure 2F, however THOR-7- mouse still seem to have somewhat expression. I am wondering how the authors can explain this, and whether their phenotype stems from reduced levels of THOR rather than a knock-out.

The authors also present a relatively high expression of THOR in the ovaries, which I think is not mentioned in the MS. Therefore I doubt that "stable deletion" is the correct wording.

For Figure 1A the authors state that they used BLAST (which is a local alignment tool), clearly for the motif detection some form of multiple alignemnt tool was used. Also I cannot see how the authors infer that THOR originated in zebra finch - it appears to me that their evolutionary phrasing is based on misconception. you could say, we THOR likely was present in the last common ancestor of zebra fish and mouse.

Also on line 20/21 p 9 the authors state "we hypothesized that THOR... altered X sperm cell development". However, I do not understand where this is based on. Do the authors investigate or know at what part of spermatogenesis such a misdevelopment might have occurred? Did you for example test the male to female sperm cell ratio? What about alternative hypothesis, such as impact on fertilisation?

Author Response

Response to Reviewer 1 Comments

Dear reviewer:

Thank you very much for giving us the opportunity to submit the revised manuscript "Investigation of the lncRNA THOR in mice highlights the importance of noncoding RNAs in mammalian male reproduction" for publication in Biomedicines. We sincerely thank you for your time and effort in thoroughly reviewing our manuscript and providing very useful comments to guide our revision. These opinions are very valuable, very helpful to the revision and improvement of our paper, and have important guiding significance for our research. We carefully studied the comments and made corrections, hoping to get approval. The revised parts are marked in red in the paper. Please find the detailed response to your comments and suggestions below, the color is red.

Point 1: I think the manuscript describes very strong phenotypes. As I am not a molecular biology expert, these in principle seems appropriate. My main questions would derive from the mthod itself. Here CRISPR/CAS9 was applied to a two exon non-coding RNA. In Figure 2F, however THOR-/- mouse still seem to have somewhat expression. I am wondering how the authors can explain this, and whether their phenotype stems from reduced levels of THOR rather than a knock-out.

Response 1: First of all, thank you very much for your approval of the article! We are very grateful for your valuable suggestions for spending your time! In Figure 2F, THOR-/- mice still have some expression, and I also found this problem in my experiment. The mouse lncRNA THOR length is 3251bp. When we use CRISPR/Cas9 to delete large fragments of the THOR, sgRNA1 is located in the upstream intron region of THOR, and sgRNA2 is located in the front of the second exon. The size of the deletion fragment is 2613bp, and the first exon region of THOR is not deleted. The qRT-PCR primer sequence for detecting THOR expression directly quoted the research published in the CELL by Hosnon et al., which is located outside the gene deletion region, and the upstream qRT-PCR primer is on the second exon of THOR. After CRISPR/Cas9 to delete the 2613b fragment of the mouse THOR gene, some fragments were still retained after the THOR gene recombination, and partial expression of THOR was still detected in qRT-PCR. In the revised manuscript, we have corrected it, we designed a new qRT-PCR primer located in the THOR deleted fragment region to verify the expression of THOR again. After the large fragment of THOR gene was deleted, THOR-/- is almost not expressed, and the result is corrected to Figure 2F. The original result is in Figure S2. It shows that after the large fragment of THOR deficient, the gene recombines to form a new truncated THOR. By predicting the centroid and MFE secondary structures (http://rna.tbi.univie.ac.at/), we can predict that the centroid and MFE secondary structures of THOR will completely change in Figure 2H. And the deletion of THOR has been verified at the gene level by PCR. Therefore, it can be considered that the phenotype of THOR-/- mice is due to the THOR deficiency.

Point 2: The authors also present a relatively high expression of THOR in the ovaries, which I think is not mentioned in the MS. Therefore I doubt that "stable deletion" is the correct wording.

Response 2: Thank you very much for your patience and careful reading of our manuscript. Our original writing caused ambiguity in the manuscript about "stable deletion". We used CRIPSPR/Cas9 to construct a mouse model of THOR gene deletion in large fragments. When detecting the THOR-deficient in female mice, we only performed genomic PCR identification to determine the genotype of THOR in female mice. "Stable deletion" originally expressed here that the deletion of the large fragment of the THOR gene in the F0 generation did not cause other mutations in the THOR gene and affect the expression of other genes in the progeny of reproduction. Does not express the meaning of "stable inheritance". In the revised manuscript, we have corrected it, Please refer to line 5 page 7 for detail.

Point 3: For Figure 1A the authors state that they used BLAST (which is a local alignment tool), clearly for the motif detection some form of multiple alignemnt tool was used. Also I cannot see how the authors infer that THOR originated in zebra finch - it appears to me that their evolutionary phrasing is based on misconception. you could say, we THOR likely was present in the last common ancestor of zebrafish and mouse.

Response 3: Thank you very much for pointing this out and letting us discover the error in the manuscript in time. We apologize for this negligence. For Figure 1A, we used BLAST alignment and speculated that THOR is highly conserved in vertebrates.The Neighbor-joining tree further showed that THOR first appeared in fish, and did not further infer that THOR first originated in zebrafish in vertebrates. Thank you very much for raising this question and correcting my mistake in presentation in time. In the revised manuscript, we have corrected it. Please refer to line 19 page 5 for detail.

Point 4: Also on line 20/21 p 9 the authors state "we hypothesized that THOR... altered X sperm cell development". However, I do not understand where this is based on. Do the authors investigate or know at what part of spermatogenesis such a misdevelopment might have occurred? Did you for example test the male to female sperm cell ratio? What about alternative hypothesis, such as impact on fertilisation?

Response 4: We gratefully appreciate for your valuable comment. Regarding the hypothesis that THOR affects the development of X sperm cells, we speculate that THOR maybe involved in the development of X sperm cells based on the imbalance of the male-male ratio of the offspring of THOR gene deletion, and we have not carried out a deeper exploration for this. Thank you very much for your valuable suggestions, In the revised manuscript, we have corrected it. Please refer to line 8 page 10 for detail.

We tried our best to improve the manuscript and made some revisions to the manuscript. We sincerely hope that this revised manuscript has resolved all your comments and suggestions. We did not list the changes here, but marked them in yellow in the revised paper. We would like to express our heartfelt thanks for your enthusiastic work and hope that the correction will be approved. Thank you very much again for your comments and suggestions.

We thank you again for taking the time to review our manuscript.

Reviewer 2 Report

Zhou et al. reported an interesting in vivo study, which tries to investigate the role of the lncRNA THOR in mouse spermatogenesis. Although the paper sounds well, it needs a substantial revision to be publishable on Biomedicines journal.

Major Concerns

-) In figure 2, Panel E should show a clearer image where PCR performed on DNA from THOR WT, THOR +/- and THOR -/- are shown. Moreover, How do the authors explain that in THOR-/- samples, THOR expression is not totally abrogated (Fig. 2F), but only reduced? The THOR abrogation is the key point of this paper.

-) In figure 3A, there is not a statistical analysis on the weights of  THOR WT, THOR +/- and THOR -/- mice. The authors should add this information.

-) Figure 3C. The authors report the survival rate of THOR WT, THOR +/- and THOR -/- mice, both male and female. This panel is to difficult to understand, the authors may separate in three different graphs, showing the Percent of Survival for male and female THOR WT, THOR +/- and THOR -/- mice. Moreover, how do the authors explain that the THOR +/- and THOR -/- female mice live shorter than WT female? Moreover, it is not reported the statistical analysis. The authors should add this information.

-) In my opinion the experiment showed in Figure 3F is not reported in a clear way. What are the graphs reported? Are they statistically significant?

-) Figure 3H is not clear. What are the n reported in figure? The figure legend does not report any info.

-) The authors state that THOR-deficient mice are smaller than WT. In my opinion, by analyzing fig 4A, this is not true. Are there any statistical analyses? Please clarify. 

-) All the figure have to be revised, organizing the panels in a clearer way (IE in the figure 3 Panel D is reported before the Panel C). Moreover, the figure legends must be improved, in order to give more experimental data to the reader.

Author Response

Response to Reviewer 2 Comments

Dear reviewer:

Thank you very much for giving us the opportunity to submit the revised manuscript "Investigation of the lncRNA THOR in mice highlights the importance of noncoding RNAs in mammalian male reproduction" for publication in Biomedicines. We sincerely thank you for your time and effort in thoroughly reviewing our manuscript and providing very useful comments to guide our revision. These opinions are very valuable, very helpful to the revision and improvement of our paper, and have important guiding significance for our research. We carefully studied the comments and made corrections, hoping to get approval. The revised parts are marked in red in the paper. Please find the detailed response to your comments and suggestions below, the color is red.

Point 1: In figure 2, Panel E should show a clearer image where PCR performed on DNA from THOR WT, THOR +/- and THOR -/- are shown. Moreover, How do the authors explain that in THOR-/- samples, THOR expression is not totally abrogated (Fig. 2F), but only reduced? The THOR abrogation is the key point of this paper.

Response 1: First of all, thank you very much for your approval of the article! We are very grateful for your valuable suggestions for spending your time! In Figure 2E, I updated the THOR gene PCR electrophoresis identification map of mice with WT, THOR+/-, and THOR-/- genotypes. In Figure 2F, THOR-/- mice still have some expression, and I also found this problem in my experiment. The mouse lncRNA THOR length is 3251bp. When we use CRISPR/Cas9 to delete large fragments of the THOR, sgRNA1 is located in the upstream intron region of THOR, and sgRNA2 is located in the front of the second exon. The size of the deletion fragment is 2613bp, and the first exon region of THOR is not deleted. The qRT-PCR primer sequence for detecting THOR expression directly quoted the research published in the CELL by Hosnon et al., which is located outside the gene deletion region, and the upstream qRT-PCR primer is on the second exon of THOR. After CRISPR/Cas9 to delete the 2613b fragment of the mouse THOR gene, some fragments were still retained after the THOR gene recombination, and partial expression of THOR was still detected in qRT-PCR. In the revised manuscript, we have corrected it, we designed a new qRT-PCR primer located in the THOR deleted fragment region to verify the expression of THOR again. After the large fragment of THOR gene was deleted, THOR-/- is almost not expressed, and the result is corrected to Figure 2F. The original result is in Figure S2. It shows that after the large fragment of THOR deficient, the gene recombines to form a new truncated THOR. By predicting the centroid and MFE secondary structures (http://rna.tbi.univie.ac.at/), we can predict that the centroid and MFE secondary structures of THOR will completely change in Figure 2H. And the deletion of THOR has been verified at the gene level by PCR. Therefore, it can be considered that the phenotype of THOR-/- mice is due to the THOR deficiency.

Point 2: In figure 3A, there is not a statistical analysis on the weights of  THOR WT, THOR +/- and THOR -/- mice. The authors should add this information.

Response 2: Thank you so much for pointing this out and we are very sorry for the negligence. In the revised manuscript, we have corrected it.

Point 3: Figure 3C. The authors report the survival rate of THOR WT, THOR +/- and THOR -/- mice, both male and female. This panel is to difficult to understand, the authors may separate in three different graphs, showing the Percent of Survival for male and female THOR WT, THOR +/- and THOR -/- mice. Moreover, how do the authors explain that the THOR +/- and THOR -/- female mice live shorter than WT female? Moreover, it is not reported the statistical analysis. The authors should add this information.

Response 3: We are very sorry for our negligence, and we totally agree with you. In the experiment, in order to determine whether THOR affects the survival of mice, we evaluated the survival of WT, THOR+/- and THOR-/- mice within 126 days. Figure 3C shows the survival curve of mice. We revised the manuscript to address your concerns and hope it is now clearer. We plot the survival curves of male WT, THOR+/- and THOR-/- mice separately from the survival curves of female WT, THOR+/- and THOR-/- mice, and the results are shown in Figure 3C.Thank you very much again for pointing this out. In the 126 days statistical survival situation, we found that the lifespan of THOR+/- and THOR-/- female mice is shorter than that of WT females. In the experiment, we haven't found the cause of the long life of female mice. This will be our next research direction to explore the role of testis-specific lncRNAs in affecting the growth and development of female animals. In the revised manuscript, we have corrected it.

Point 4: In my opinion the experiment showed in Figure 3F is not reported in a clear way. What are the graphs reported? Are they statistically significant?

Response 4: Thank you very much for pointing this out and we are very sorry for the negligence. In order to evaluate the influence of THOR on embryonic development in vitro, we microinject a mixture of sgRNAs and Cas9 mRNA into fertilized eggs and delete large fragments of the THOR gene. Every 100 embryos are used as a group, and 3 groups are repeated. Count the number of embryos developed into 2-cell, 4-cell, morula and blastocysts in vitro, and calculate the development rate. We once again apologize for our negligence. In the revised manuscript, we have corrected it.

Point 5: Figure 3H is not clear. What are the n reported in figure? The figure legend does not report any info.

Response 5: Thank you very much for pointing this out and we are very sorry for the negligence. Here, we are continuing the experiment in Figure 3G. After 9 male and female mice of different genotypes mate, check the vaginal suppository the next morning. The embryos of female mice with vaginal suppositories were taken out and cultured in vitro, and the number of embryos was counted. After culturing the embryos in vitro for 24 hours, observe the development of 2-cell, and count the number of 2-cells as the fertilization ability of the corresponding male mice. In the figure, n is the total number of embryos in the mother mouse whose vaginal suppository was detected in Figure 3G. I am very sorry for the confusion, In the revised manuscript, we have corrected it.

Point 6: The authors state that THOR-deficient mice are smaller than WT. In my opinion, by analyzing fig 4A, this is not true. Are there any statistical analyses? Please clarify.

Response 6: Thank you very much for your comments, and we apologize for our negligence. In Figure 4A, in the process of taking pictures of mice, in order to more clearly compare that THOR-/- mice are smaller than WT and THOR+/-, so we put THOR-/- mice in the middle position. Because the previous mark was relatively small, it caused your doubts. We have made the corresponding changes. Regarding the size of the mice, we only carried out the statistics of body weight, as shown in Figure 3A. In the revised manuscript, we have corrected it.

Point 7: All the figure have to be revised, organizing the panels in a clearer way (IE in the figure 3 Panel D is reported before the Panel C). Moreover, the figure legends must be improved, in order to give more experimental data to the reader.

Response 7: Thank you very much for your careful inspection and your valuable comments. We completely agree with your comments. Once again, we apologize for your negligence. We have modified all the drawings and re-typesetting.

We tried our best to improve the manuscript and made some revisions to the manuscript. We sincerely hope that this revised manuscript has resolved all your comments and suggestions. We did not list the changes here, but marked them in yellow in the revised paper. We would like to express our heartfelt thanks for your enthusiastic work and hope that the correction will be approved. Thank you very much again for your comments and suggestions.

We thank you again for taking the time to review our manuscript.

Round 2

Reviewer 1 Report

Thanks for replying to my comments

Reviewer 2 Report

The authors have improved the paper. It is now ready for the publication